# Wisdom of the Ensemble: Improving Consistency of Deep Learning Models

**Lijing Wang**
University of Virignia
lw8bn@virginia.edu

**Dipanjan Ghosh**
Hitachi America Ltd.
dipanjan.ghosh@hal.hitachi.com

**Maria Teresa Gonzalez Diaz**
Hitachi America Ltd.
teresa.gonzalezdiaz@hal.hitachi.com

**Ahmed Farahat**
Hitachi America Ltd.
ahmed.farahat@hal.hitachi.com

**Mahbubul Alam**
Hitachi America Ltd.
Mahbubul.Alam@hal.hitachi.com

**Chetan Gupta**
Hitachi America Ltd.
chetan.gupta@hal.hitachi.com

**Jiangzhuo Chen**
University of Virignia
chenj@virginia.edu

**Madhav Marathe**
University of Virignia
marathe@virginia.edu

## Abstract

Deep learning classifiers are assisting humans in making decisions and hence the user's trust in these models is of paramount importance. Trust is often a function of constant behavior. From an AI model perspective it means given the same input the user would expect the same output, especially for correct outputs, or in other words consistently correct outputs. This paper studies a model behavior in the context of periodic retraining of deployed models where the outputs from successive generations of the models might not agree on the correct labels assigned to the same input. We formally define *consistency* and *correct-consistency* of a learning model. We prove that consistency and correct-consistency of an ensemble learner is not less than the average consistency and correct-consistency of individual learners and correct-consistency can be improved with a probability by combining learners with accuracy not less than the average accuracy of ensemble component learners. To validate the theory using three datasets and two state-of-the-art deep learning classifiers we also propose an efficient dynamic snapshot ensemble method and demonstrate its value. Code for our algorithm is available at https://github.com/christa60/dynens.

## 1 Introduction

As AI is increasingly supporting humans in decision making [34, 13], there is more emphasis than ever on building trustworthy AI systems [5]. Despite the discrepancies in terminology, almost every recent research on trustworthy AI agree on the need for building AI models that produce consistently correct outputs for the same input [14]. Although this seems like a straightforward requirement, however as we periodically retrain AI models in the field, there is no guarantee that different generations of the model will be consistently correct when presented with the same input. Consider an example of an AI-based car-safety system [11] that correctly detects distracted driving and then does not detect

for the same input a day later after being retrained overnight. Attackers can easily sport such an inconsistent behavior and exploit them, further this can compromise safety of drivers. Also consider the damage that can be caused by an AI-agent for COVID-19 diagnosis that correctly recommends a true patient to self-isolate, and subsequently changes its recommendation after being retrained with more data. Regression models like in Remaining Useful Life estimation system also suffer from inconsistency problem. However, considering that classification models are more predominant we focus on such models only.

In this paper, we define *consistency* of a model as the ability to make consistent predictions across successive model generations for the same input. This definition is different from the replicability of model performance at an aggregate level [12] - with a stable training pipeline of a classifier, aggregate metrics can be relatively consistent across successive generations, but changes to the training data or even retraining on the same data often causes changes in the individual predictions. Consistency is applicable to both correct and incorrect outputs, however, the more desirable case is producing consistently correct outputs for the same inputs. We define ability to make consistent correct predictions across successive model generations for the same input as *correct-consistency*.

To understand further the effect of consistency and correct-consistency on users' trust, lets consider the following scenarios for the car safety example (driver distraction) with two model generations - $Model_i$ and $Model_{i+1}$, and an input $X$. (1) If the outputs from both models are correct, then the issue of inconsistency does not arise and does not affect the user. This is a case of correct-consistency. (2) If the output from $Model_i$ is incorrect while from $Model_{i+1}$ is correct, it won't adversely affect users' trust but in-fact can be considered as an improvement in the system. (3) If the output from $Model_i$ is correct while from $Model_{i+1}$ is incorrect, it is a very severe case because this can adversely affect users trust in the system as well as its usability. In this case correct-consistency is desired. (4) If the outputs from both models are incorrect, although it is an undesirable scenario and can affect users but it is still less severe from consistency point of view.

Although mentioned in limited studies [4, 28, 20] no previous work has discussed or measured consistency formally. In this work, we investigate why and how ensembles can improve *consistency* and *correct-consistency* of deep learning classifiers theoretically and empirically. Ensembles have had success in improving accuracy and uncertainty quantification [24, 32], but have not been studied in the context of consistency. To the best of our knowledge, we are the first to define and measure the consistency for deep learning classifiers, and improve it using ensembles. Specifically we make the following contributions:

- Formally define *consistency* and *correct-consistency* of a model and multiple metrics to measure it;
- Provide a theoretical explanation of why and how ensemble learning can improve consistency and correct-consistency when the average performance of all predictors in the ensemble is considered;
- Prove that the consistency and correct-consistency of an ensemble learner is not less than the average consistency and average correct-consistency of individual learners;
- Prove that adding components with accuracy higher than the average accuracy of ensemble component learners to an ensemble learner can yield a better consistency for correct predictions;
- Propose a dynamic snapshot ensemble learning with pruning algorithm to boost predictive correct-consistency and accuracy in an efficient way;
- Conduct experiments on CIFAR10, CIFAR100, and Yahoo!Answers using state-of-the-art deep learning classifiers and demonstrate effectiveness and efficiency of the proposed method and prove the theorems empirically.

## 2 Related work

**Reproducibility and consistency**: Traditionally, reproducibility refers to the ability to replicate a scientific study [27, 22] or reproduce the model performance at an aggregate level [20, 28]. Kenett et.al. in [22] suggest to clarify the terminology of reproducibility, repeatability and replicability by considering the intended generalization of the study. Consistency as we define is the ability to reproduce the same predictions across different generations of a model for the same input. There are only a limited number of studies related to consistency of deep learning classifiers [4, 28, 20]. Among

these works, only Anil et.al. in [4] state that their proposed distillation method can reproduce the same predictions and they measure the mean absolute difference between the predictions of retrained models. However, their work did not focus on measuring the consistency. Patil et.al. [28], Islam et.al. [20] speculate that ensembles can improve consistency but with no followup investigation. To the best of our knowledge, this is the first paper to define, investigate, address and validate consistency formally.

**Ensemble in deep learning**: Ensemble methods have been widely used in machine learning leading to an improvement in accuracy. Ren et. al. [31] present a comprehensive review. Deep ensembles, have been successful in boosting predictive performance [33, 1, 10, 3, 35, 19], as well as in estimating predictive uncertainty [16, 24, 32]. Wang et.al. [33], Pietruczuk et.al. [29] and Bonab et.al. [7] explore the ensemble size, diversity, and efficiency of deep ensembles. In this paper, the deep ensembles for classification are categorized into two classes: (1) *model-independent* methods where ensemble techniques can be applied directly to any base models like bagging [8] and boosting [15], dropout [18], and snapshot ensemble [19]; (2) *model-dependent* methods where ensemble techniques are proposed for the context of specific models like deep SVM [1], multi-column CNN [9], multi-column Autoencoder [3]. We focus on model-independent methods as they are suited in the context of post-deployment. Conventional ensembles like bagging and boosting can be extended to deep learning classifiers. Lakshminarayan et.al. [24] and Lee et.al. [25] demonstrate that training on entire dataset with random shuffling and with random initialization of the neural network parameters is better than bagging for deep ensembles, that we call as extended bagging. Baldi et.al. [6] discuss that dropout (proposed by Hinton et.al. [18]) can be seen as an extreme form of bagging in which each model is trained on a single case and each parameter of the model is regularized by sharing it with the corresponding parameter in all the other models. Gal et.al. [16] prove that using dropout technique is equivalent to Bayesian NN's and propose Monte Carlo Dropout (MC Dropout) to estimate uncertainty in deep learning. Huang et.al. in [19] propose the snapshot ensemble method that trains a neural network to converge to multiple local optima along its optimization path by using cyclic learning rate schedules [26] to create an ensemble, consistently yielding lower error rates than state-of-the-art single models at no additional training cost.

The above efforts are mainly directed towards improving accuracy and uncertainty quantification. In this work, we use ensembles to investigate and address consistency of deep learning classifiers, with the objective to improve correct-consistency in the context of re-training. We provide a theoretical explanation of why and how ensemble learning can improve consistency and correct-consistency and experimentally validate it on several datasets and state-of-the-art deep classifiers by using a new dynamic snapshot ensemble method that combines extended bagging and snapshot techniques with a dynamic pruning algorithm. The method is efficient and feasible for post deployment of a model. It should be noted that combining classifiers may not necessarily be better than the consistency performance of the best classifier in the ensemble. This is a well-known issue when using ensembles to improve other performance metrics (e.g. accuracy) and has been discussed in previous research [30]. We hope that the simplicity and strong empirical performance of our approach will spark more interest in deep learning for consistency estimation.

## 3   Problem definition

We use the term *consistency*, *correct-consistency* and the following definitions in this work:

**Definition 1.** *Model:* An architecture built for a learning task.

**Definition 2.** *Trained learner:* A predictor (single or ensemble) of a model after a training cycle. A model can have multiple trained learners as a result of multiple training cycles.

**Definition 3.** *Copy of a trained learner:* Two trained learners generated by re-training the model with the same training process settings on the same or different training datasets.

**Definition 4.** *Consistency of a model:* The ability of the model to reproduce an output for the same input for multiple trained learners, irrespective of whether the outputs are correct/incorrect.

**Definition 5.** *Correct-consistency of a model:* The ability of the model to reproduce a correct output for the same input for multiple trained learners.

Given that the training dataset increases as new online data streams come in, i.e. $D_1 \subseteq D_2 \subseteq \cdots \subseteq D_T$, a model $M$ when trained on $D_i$ results in a trained learner $L_i$. Given a testing dataset

$I$, each trained learner $L_i$ makes prediction $\hat{Y}_i$ ($i \in \{1, \cdots, T\}$) for $I$. $\hat{Y}_i$ is a set of individual one-hot encoding prediction vectors. The overlap between any two predictions $\hat{Y}_i$ and $\hat{Y}_j$ is used to measure the consistency of $M$, while overlap between any two correct predictions is used to measure correct-consistency. Trained learners $L_i$ can thus have the same accuracy but different overlap between predictions $\hat{Y}_i$ and $\hat{Y}_j$. Our objective is to improve consistency as well as correct-consistency for deep learning classification.

## 4 Why ensemble?

Ensemble learning methods combine the predictions from multiple trained learners to reduce the variance of predictions and reduce generalization error. Our intuition and premise of using ensemble is to leverage the models from local optima, to obtain greater coverage of the feature space, get consensus for the predictions and then produce the final output.

Assume that a supervised classification problem has $p$ class labels, $C = \{C_1, \ldots, C_p\}$. Consider an ensemble of $m$ component single trained learners, $\xi = \{SL_1, \ldots, SL_m\}$, and $n$ testing data points, $I = \{I_1, \ldots, I_n\}$. For a data point $I_t (1 \le t \le n)$, a single trained learner $SL_j (1 \le j \le m)$ outputs a prediction vector, $s_{tj} = \langle S_{tj}^1, \ldots, S_{tj}^p \rangle$ where $\sum_{k=1}^{p} S_{tj}^k = 1$. The prediction vectors from $m$ component learners are combined using a weight vector $w = \langle W_1, \ldots, W_m \rangle$, $\zeta = f(w, \xi)$, where $W_j$ is the weight for $SL_j$. The copy of $SL_j$ is denoted as $\tilde{SL}_j$, similarly for $\tilde{\xi}$, $\tilde{\zeta}$. The true label vector for $I_t$ is denoted as $r_t = \langle R_t^1, \ldots, R_t^p \rangle$. The complete list of notations and symbols are presented in Appendix A.

We represent the prediction and ground truth vectors in a $p$-dimensional space where they belong to the $(p-1)$ dimensional probability simplex. The consistency of a prediction is represented as the Euclidean distance between two prediction vectors in $p$-dimensional space [7], the distance between $s_{tj}$ and $\tilde{s}_{tj}$ is denoted as:

$$distance(s_{tj}, \tilde{s}_{tj}) = \sqrt{\sum_{k=1}^{p} (S_{tj}^k - \tilde{S}_{tj}^k)^2} \qquad (1)$$

And the correct-consistency of a prediction is represented as the sum of Euclidean distance between the ground truth vector and two prediction vectors, denoted as:

$$distance(s_{tj}, \tilde{s}_{tj}, r_t) = distance(s_{tj}, \tilde{s}_{tj}) + distance(s_{tj}, r_t) + distance(\tilde{s}_{tj}, r_t) \qquad (2)$$

*A smaller distance corresponds to a higher consistency/correct-consistency and a higher consistency/correct-consistency is better.* There are other supervised problems where this statement is not true, e.g. multi-label classification, which is not in scope of this work.

If we use *averaging* as the output combination method for $\zeta$, i.e. $W_j = 1$ for all $j (1 \le j \le m)$, the final prediction vector for $\zeta$ is represented by the centroid-point of the prediction vectors of all single learners in $\zeta$. Thus, for a given data point $I_t$, there is a mapping to the centroid-point vector, $o_t = \langle O_t^1, \ldots, O_t^p \rangle$ where,

$$O_t^k = \frac{1}{m} \sum_{j=1}^{m} S_{tj}^k (1 \le k \le p) \qquad (3)$$

**Theorem 1.** For $I_t$, the distance between the centroid-vectors $o_t$ and $\tilde{o}_t$ is not greater than the average distance between a pair of prediction vectors $(s_{tj}, \tilde{s}_{tj})$ of $m$ component learners.

$$distance(o_t, \tilde{o}_t) \le \frac{1}{m} \sum_{j=1}^{m} distance(s_{tj}, \tilde{s}_{tj}) \qquad (4)$$

For proof, refer Appendix B.

**Theorem 2.** For $I_t$, let $\xi_l = \xi - SL_l (1 \le l \le m)$ be a subset of ensemble $\xi$ without $SL_l$. If each $\xi_l$ has $o_{tl}$ as its centroid-vector. Then,

$$distance(o_t, \tilde{o}_t) \le \frac{1}{m} \sum_{l=1}^{m} distance(o_{tl}, \tilde{o}_{tl}) \qquad (5)$$

For proof, refer Appendix C. The upper bound for $distance(o_t, \tilde{o}_t)$ is determined by $\frac{1}{m} \sum_{j=1}^{m} distance(s_{tj}, \tilde{s}_{tj})$, while the lower bound is 0.

**Theorem 3.** For $I_t$, the sum of distances between the centroid-vectors and ground truth vector $(o_t, \tilde{o}_t, r_t)$ is not greater than the average distance between the prediction vectors and ground truth vector $(s_{tj}, \tilde{s}_{tj}, r_t)$ of $m$ component learners.

$$distance(o_t, \tilde{o}_t, r_t) \leq \frac{1}{m} \sum_{j=1}^{m} distance(s_{tj}, \tilde{s}_{tj}, r_t) \tag{6}$$

For proof, refer Appendix D.

**Theorem 4.** For $I_t$, let $\xi_l = \xi - SL_l (1 \leq l \leq m)$ be a subset of ensemble $\xi$ without $SL_l$. If each $\xi_l$ has $o_{tl}$ as its centroid-vector. Then,

$$distance(o_t, \tilde{o}_t, r_t) \leq \frac{1}{m} \sum_{l=1}^{m} distance(o_{tl}, \tilde{o}_{tl}, r_t) \tag{7}$$

For proof, refer Appendix D. The upper bound for $distance(o_t, \tilde{o}_t, r_t)$ is determined by $\frac{1}{m} \sum_{j=1}^{m} distance(s_{tj}, \tilde{s}_{tj}, r_t)$, while the lower bound is 0.

Let $acc_\zeta = \frac{1}{n} \sum_{t=1}^{n} 1_{\zeta,r}^{1}(t)$ and $acc_{\tilde{\zeta}} = \frac{1}{n} \sum_{t=1}^{n} 1_{\tilde{\zeta},r}^{1}(t)$ denote the prediction accuracy of $\zeta$ and $\tilde{\zeta}$ for $I$, $ccon(\zeta, \tilde{\zeta}) = \frac{1}{n} \sum_{t=1}^{n} 1_{\zeta,\tilde{\zeta},r}(t)$ denotes the correct-consistency between $\zeta$ and $\tilde{\zeta}$, where $1_{\zeta,r}^{1}(t)$ and $1_{\zeta,\tilde{\zeta},r}^{1}(t)$ are indicator functions defined by Eq. 17 and 15 in Appendix H. We have

**Theorem 5.** For $I$, the correct-consistency between two learners $ccon(\zeta, \tilde{\zeta})$ is not greater than the smaller of the two accuracy $acc_\zeta$ and $acc_{\tilde{\zeta}}$, and is no less than the minimum overlap between $acc_\zeta$ and $acc_{\tilde{\zeta}}$.

$$max(acc_\zeta + acc_{\tilde{\zeta}} - 1, 0) \leq ccon(\zeta, \tilde{\zeta}) \leq min(acc_\zeta, acc_{\tilde{\zeta}}) \tag{8}$$

For proof, refer Appendix F

**Corollary 5.1.** For $I$, let $\zeta_l$ and $\tilde{\zeta}_l$ be the ensemble functions of $\xi_l$ and $\tilde{\xi}_l$. If $acc_{SL_l}$ is greater or equal to $\frac{1}{m-1} \sum_{SL_i \in \xi_l} acc_{SL_i}$, then, at least with a probability $\rho$ that

$$ccon(\zeta_l, \tilde{\zeta}_l) \leq ccon(\zeta, \tilde{\zeta}) \tag{9}$$

where $\rho$ is quantifiable by $acc_\zeta, acc_{\tilde{\zeta}}, acc_{\zeta_l}, acc_{\tilde{\zeta}_l}$. For proof, refer Appendix G.

**Discussion**: Theorem 1 shows that the consistency of an ensemble model is higher or equal to the average consistency of all individual component learners. Theorem 2 can be generalized for any subset of $\xi$ with $m - d$ $(1 \leq d \leq m - 1)$ component learners. It shows that the consistency of an ensemble model with $m$ component learners is higher or equal to the average consistency of ensembles with $m - d$ component learners. Thus a higher ensemble consistency can be achieved by combining more components with small variance in predictions, however, this cannot guarantee a better consistency in correct predictions which is more desirable. Similarly, Theorem 3 and 4 show that a higher correct-consistency can be achieved by combining more components with small variance and error in predictions. Theorem 5 and Corollary 5.1 show that a better aggregate correct-consistency performance of an ensemble can be achieved by combining components with accuracy that is higher than the average accuracy of the ensemble members. Theorems 1 and 2 can be generalized to other distance metrics using Minkowski distance (refer Appendix E for proofs). The theorems are not limited to classification problems, and are applicable for regression problems by considering $p = 1$ and $s_{tj} = \langle S_{tj} \rangle$ as prediction value, where $S_{tj} \in \mathbb{R}$, however we limit this paper to classification tasks only. Further, all theoretical findings are invariant with respect to changes in training data distribution over successive model generations.

## 5 Dynamic snapshot ensemble method

We propose dynamic snapshot ensemble (DynSnap) method by combining extended bagging [24] like random initialization of model parameters and random shuffle of the training dataset; snapshot

ensemble [19] techniques with a dynamic pruning algorithm. Algorithm 1 outlines the procedure of generating a snapshot ensemble learner using dynamic pruning. The details of the techniques are presented below.

---

**Algorithm 1:** Pseudocode of the dynamic snapshot ensemble (DynSnap)

---

**Input:** $M$: a model of classification problem; $D$: training dataset; $m$: number of ensemble single learners; $N$: number of snapshot trained learners from one training process; $\beta$: prune factor

**Output:** $\zeta$: ensemble learner

1   $\xi \leftarrow \emptyset$ // The set of ensemble components
2   $w \leftarrow \emptyset$ // The set of weights of ensemble components
3   **while** *len($\xi$) < m* **do**
4     Resample training and validation dataset $TD_i$ from $D$. Train $M$ on $TD_i$ using snapshot learning and save trained learners $\xi_i = \{SL_{i1}, \ldots, SL_{iN}\}$. Save validation accuracy $w_i = \{W_{i1}, \ldots, W_{iN}\}$ for $\xi_i$. Sort $\xi_i$ in descending order based on $w_i$.
5     $a := max(w_i), b := min(w_i)$
     /* Start pruning                                                           */
6     **for** *j in* $\{1, 2, \ldots, N\}$ **do**
7       **if** $W_{ij} \geq (1 - \beta) * a + \beta * b$ **then**
8         $\xi \cup SL_{ij}, w \cup W_{ij}$
9         **if** $len(\xi) \geq m$ **then**
10           break

11   $\zeta := \mathcal{F}(\xi, w)$ // $\mathcal{F}$ is combination method including MV,WMV,AVG,WAVG

---

**Snapshot learning**: In snapshot ensemble learning [19], instead of training $N$ neural networks independently, the optimizer converges $N$ times to local optima along its optimization path, thereby training multiple single learners at no additional cost. We extend the snapshot learning in two dimensions - learning rate schedule (cyclic annealing schedule and step-wise decay schedule) and snapshot saving strategy (cyclic snapshot and top-N snapshot). In *cyclic annealing schedule* [26] the learning rate $\alpha$ is updated to follow a cosine function $\alpha(t) = \frac{\alpha_0}{2}(\cos(\frac{\pi \mathrm{mod}(t-1, \lceil T/N \rceil)}{\lceil T/N \rceil}) + 1)$ where $\alpha_0$ is the initial learning rate, $t$ is the iteration or epoch number, $T$ is the total iteration or epoch number and $\alpha(t)$ is the learning rate at time point $t$. In *step-wise decay schedule* the learning rate decays with a decay factor every epoch by $\alpha(t) = F_d(t)\alpha(t - 1)$ where $t$ is the epoch number, $F_d(t)$ is the decay factor that is a function of epoch number. For snapshot saving strategy, that is a method to select and save $N$ single trained learners during one training cycle, we leverage the *cyclic snapshot* where a best single trained learner is saved periodically every $\lceil T/N \rceil$ iterations or epochs based on the validation accuracy, and *top-N snapshot* where top $N$ single trained learners during model training are saved based on validation accuracy.

We implement two snapshot learning methods: ***DynSnap-cyc*** uses cyclic annealing schedule and cyclic snapshot strategy with $t$ as epoch number; ***DynSnap-step*** uses step-wise decay schedule and top-N snapshot strategy. Note that in the proposed methods, the learning rate schedule can be any appropriate schedule as long as the cyclic schedules use cyclic snapshot while the other decay schedules (e.g. exponential decay or no decay) use top-N snapshot. We choose cosine function and step-wise decay as they are classic and have been extensively validated by previous works.

**Dynamic ensemble pruning**: Given a model $M$ and a training dataset $D$, our goal is to include $m$ single learners with good accuracy in the final ensemble learner $\zeta$. We propose a pruning algorithm that filters good single learners locally according to a pruning criteria and combines local optimal learners across multiple learning processes by random initialization on model parameters and random shuffling on training and validation datasets. The proposed pruning algorithm conforms with Corollary 5.1 and creates data diversity in $\zeta$.

For a single snapshot learning on a resampled training dataset $TD_i$, $N$ single learners are snapshotted, $\xi_i = \{SL_{i1}, \ldots, SL_{iN}\}$, with their validation accuracy, $w_i = \{W_{i1}, \ldots, W_{iN}\}$. The **pruning criteria** $\mathcal{P}$ is defined as: $SL_{ij}$ is included in $\zeta$ if $W_{ij}$ is larger or equal to a threshold $\tau$:

$$\tau = (1 - \beta) * max(w_i) + \beta * min(w_i) \tag{10}$$

where $max(w_i)$, $min(w_i)$ are the maximum and minimum weights of $w_i$. According to Theorem 5 and Corollary 5.1, $\tau = \frac{1}{N} \sum w_i$ i.e. $\beta = \frac{max(w_i) - \frac{1}{N} \sum w_i}{max(w_i) - min(w_i)}$ selects $SL_{ij}$ that can leads to better correct-consistency of the ensemble than the correct-consistency of $\xi_i$, resulting into an ideal $\beta$ for $\xi_i$ empirically. Note that the pruning algorithm does not guarantee a better correct-consistency because $w_i$ is the validation accuracy which is an estimation of the testing accuracy.

By pruning on $\xi_i$, $N_i$ $(1 \leq N_i \leq N)$ trained learners are combined to form an ensemble $\zeta$. The training and pruning is repeated with random initialization on model parameters along with random shuffling of training and validation datasets until the number of trained learners in $\zeta$ is $m$, i.e. $\sum_{i=1}^{d} N_i \geq m$. The set of ideal $\beta$s is denoted as $\beta^*$, that is non-user specified and is the default setting for DynSnap-cyc and DynSnap-step. However, by manually varying $\beta$ we aim to validate Corollary 5.1 experimentally using a sensitivity analysis shown in Section 6.

**Combination methods**: In our method, an output combination method $\mathcal{F}$ is used for ensemble trained learners $\zeta = \mathcal{F}(w, \xi)$, where $\zeta$ is the final ensemble learner, $\xi$ is the final set of trained learners, $w$ is the corresponding weight vector. $\mathcal{F}$ can be majority voting (MV) on predicted one-hot vectors or averaging (AVG) on predicted score vectors or weighted voting (WMV) or weighted averaging (WAVG). For the latter two, the weight vector $w = \langle W_1, \ldots, W_m \rangle$ is the validation accuracy of $m$ single learners saved during training processes.

# 6 Experimental results and analysis

We conduct experiments on three datasets and compare with two state-of-the-art model-independent deep ensemble method to validate our theorems by evaluating consistency and correct-consistency using defined metrics and to analyze the advantage of our proposed method. In our experiments, a deep classifier is chosen as the base model for each dataset. All deep ensemble methods are implemented on the top of the base model.

**Metrics**: With two learners $L_i$ and $L_j$, $\hat{Y}_i$ and $\hat{Y}_j$ as corresponding predictions for testing data $I$, assuming that $A \subseteq \hat{Y}_i$ and $B \subseteq \hat{Y}_j$ are correct prediction sets, then the correct predictions overlap is $A \cap B$. For testing data set $I$, metrics used to evaluate accuracy, consistency and correct-consistency are: (1) Consistency (CON): $\frac{|\hat{Y}_i \cap \hat{Y}_j|}{n}$. (2) Accuracy (ACC): $\frac{|A|}{n} or \frac{|B|}{n}$. (3) Correct-Consistency (ACC-CON): $\frac{|A \cap B|}{n}$, where $|\cdot|$ is the size of a set. Additional metrics and details are presented in Appendix H. Results for additional metrics are presented in Appendix K

**Data and models**: We conduct experiments using three datasets and two state-of-the-art models. *YAHOO!Answers* [36] is a topic classification dataset with 10 output categories, 140K and 6K training and testing samples. *CIFAR10* and *CIFAR100* [23] are datasets with 10 and 100 output categories respectively, 50k and 10k color images as training and testing samples. We use *ResNet* [17] for CIFAR10 and CIFAR100 and *fastText* [21] for for YAHOO!Answers. To simulate online data streams with imbalanced class distribution, we reorganize each dataset so that three class imbalanced training sets $D_1 \subseteq D_2 \subseteq D_3$ are generated for each dataset. The dataset, models and hyper-parameters are shown in Table 1. Details of the datasets and how we generate them are presented in Appendix J.

Table 1: Data and Models

|  | CIFAR10 | | | CIFAR100 | | | YAHOO!Answers | | |
|---|---|---|---|---|---|---|---|---|---|
| **Data** | $D_1$ | $D_2$ | $D_3$ | $D_1$ | $D_2$ | $D_3$ | $D_1$ | $D_2$ | $D_3$ |
| Classes | 9 | 9 | 10 | 99 | 99 | 100 | 9 | 9 | 10 |
| Training | 24.4K | 30.5K | 32.5K | 37.5K | 46.9K | 47.4K | 683.2K | 768.6K | 910K |
| Validation | 4500 | 4500 | 4500 | 4950 | 4950 | 4950 | 10K | 10K | 10K |
| Testing | 4500 | 4500 | 4500 | 4950 | 4950 | 4950 | 50K | 50K | 50K |
| **Model** | **ResNet20** | | | **ResNet56** | | | **fastText** | | |
| Epochs | 200 | | | 200 | | | 200 | | |
| Initial-lr | $1e-03$ | | | $1e-03$ | | | $1e-03$ | | |

**Methods for comparison**: Given a base model $M$, **SingleBase** is a single learner using original learning procedure. **ExtBagging** combines $m$ single learners using original learning procedure with random initialization and random shuffle of training dataset. **MCDropout** [16] adds a dropout layer to the base model and train a single learner which predicts $m$ times during inference time. **Snapshot** [19] combines $m$ single learners from DynSnap-cyc learning without pruning ($\beta = 1$). **DynSnap-**

*cyc* combines $m$ single learners from DynSnap-cyc learning with dynamic pruning. ***DynSnap-step*** combines $m$ single learners from DynSnap-step learning with dynamic pruning.

**Experiment settings**: The experiment settings for SingleBase models are shown in Table 1. We set $m = 20$ for ensemble methods, and $N = 10$, $\beta = \beta^*$ for DynSnap-cyc and DynSnap-step, $F_d(t)$ in DynSnap-step is $1e - 1$, $1e - 2$, $1e - 3$ at 80, 120, 160 epochs, dropout with 0.1 drop probability. We extend epoch number to 400 for Snapshot to get 20 single learners from one training. The accuracy (ACC) reported is the average of three individual accuracies on $D_i$, while consistency (CON) and correct-consistency (ACC-CON) are the average of three values computed from any two of $D_i$. The final results are the average of 5 replicates per method. All datasets are reorganized so that they have imbalanced classes, hence the SingleBase performance is not comparable with the state-of-the-art performances as the training and testing sets are different. However, to make sure that the implementation of the models is correct, we did conduct experiments to replicate the state-of-the-art performance with a $\pm 2\%$ margin of error. For brevity we are not reporting these replication results. We report results for four combination methods $\zeta$ used - MV, WMV, AVG, and WAVG. Validation accuracy is used as the weight of a single learner for WMV and WAG.

Table 2: ACC, CON, ACC-CON performance and run-time of each method. $\kappa$ is one training time from scratch using the base model and model settings. Bold face the best performance and underline the second-best.

| | CIFAR10+ResNet20 | | | | CIFAR100+ResNet56 | | | | YAHOO!Answers+fastText | | | |
|---|---|---|---|---|---|---|---|---|---|---|---|---|
| **ACC**(%) | MV | WMV | AVG | WAVG | MV | WMV | AVG | WAVG | MV | WMV | AVG | WAVG |
| SingleBase | 85.84 | 85.84 | 85.84 | 85.84 | 67.60 | 67.60 | 67.60 | 67.60 | 63.70 | 63.70 | 63.70 | 63.70 |
| ExtBagging | 88.46 | **88.76** | **89.14** | **89.14** | 75.25 | 75.27 | 75.60 | 75.61 | 65.61 | **65.82** | **65.64** | **65.64** |
| MCDropout | 85.41 | 85.41 | 85.41 | 85.41 | 67.47 | 67.47 | 67.47 | 67.47 | 64.13 | 64.13 | 64.13 | 64.13 |
| Snapshot | 87.48 | 87.59 | 87.66 | 87.67 | 72.60 | 72.56 | 72.96 | 72.96 | 64.37 | 64.38 | 64.35 | 64.35 |
| DynSnap-cyc | 88.30 | 88.36 | 88.47 | 88.47 | 75.40 | 75.45 | 75.64 | 75.64 | 64.97 | 64.98 | 65.01 | 65.01 |
| DynSnap-step | **88.50** | 88.50 | 88.87 | 88.87 | 73.14 | 73.20 | 74.12 | 74.11 | 64.90 | 64.89 | 64.95 | 64.95 |
| **CON**(%) | MV | WMV | AVG | WAVG | MV | WMV | AVG | WAVG | MV | WMV | AVG | WAVG |
| SingleBase | 85.38 | 85.38 | 85.38 | 85.38 | 67.19 | 67.19 | 67.19 | 67.19 | 88.46 | 88.46 | 88.46 | 88.46 |
| ExtBagging | 92.01 | 92.03 | 92.84 | 92.83 | 84.37 | 84.40 | 85.68 | 85.69 | 92.56 | 92.36 | 92.81 | 92.83 |
| MCDropout | 85.30 | 85.30 | 85.30 | 85.30 | 67.08 | 67.08 | 67.08 | 67.08 | 85.01 | 85.01 | 85.01 | 85.01 |
| Snapshot | 89.79 | 89.81 | 89.88 | 89.89 | 76.65 | 76.49 | 76.94 | 76.98 | 91.72 | 91.76 | 91.87 | 91.87 |
| DynSnap-cyc | **92.63** | **92.56** | **93.04** | **93.05** | **85.04** | **85.15** | **85.72** | **85.70** | **92.72** | **92.72** | **92.88** | **92.88** |
| DynSnap-step | 91.64 | 91.48 | 92.29 | 92.30 | 78.26 | 78.39 | 81.23 | 81.16 | 92.32 | 92.29 | 92.41 | 92.41 |
| **ACC-CON**(%) | MV | WMV | AVG | WAVG | MV | WMV | AVG | WAVG | MV | WMV | AVG | WAVG |
| SingleBase | 79.86 | 79.86 | 79.86 | 79.86 | 57.80 | 57.80 | 57.80 | 57.80 | 60.35 | 60.35 | 60.35 | 60.35 |
| ExtBagging | 85.21 | 85.46 | **86.12** | **86.12** | 70.53 | 70.61 | 71.25 | 71.26 | 62.52 | 62.65 | 62.65 | 62.65 |
| MCDropout | 79.37 | 79.37 | 79.37 | 79.37 | 57.72 | 57.72 | 57.72 | 57.72 | 59.27 | 59.27 | 59.27 | 59.27 |
| Snapshot | 83.37 | 83.47 | 83.53 | 83.53 | 65.70 | 65.60 | 66.14 | 66.13 | 61.71 | 61.73 | 61.75 | 61.74 |
| DynSnap-cyc | **85.27** | 85.33 | 85.60 | 85.61 | **70.89** | **71.00** | **71.37** | **71.36** | **62.65** | **62.66** | **62.74** | **62.74** |
| DynSnap-step | 85.08 | 85.01 | 85.67 | 85.68 | 66.86 | 66.97 | 68.51 | 68.51 | 62.44 | 62.43 | 62.53 | 62.53 |
| **Run-time**($\kappa$) | CIFAR10+ResNet20 | | | | CIFAR100+ResNet56 | | | | YAHOO!Answers+fastText | | | |
| SingleBase | $\kappa_1$ | | | | $\kappa_2$ | | | | $\kappa_3$ | | | |
| ExtBagging | $20 * \kappa_1$ | | | | $20 * \kappa_2$ | | | | $20 * \kappa_3$ | | | |
| MCDropout | $\kappa_1$ | | | | $\kappa_2$ | | | | $\kappa_3$ | | | |
| Snapshot | $2 * \kappa_1$ | | | | $2 * \kappa_2$ | | | | $2 * \kappa_3$ | | | |
| DynSnap-cyc | $3 * \kappa_1$ | | | | $4 * \kappa_2$ | | | | $5.3 * \kappa_3$ | | | |
| DynSnap-step | $7 * \kappa_1$ | | | | $5.3 * \kappa_2$ | | | | $5.3 * \kappa_3$ | | | |

**Numerical results and analysis**: The performance results for all datasets, methods and metrics are shown in Table 2. In general, the results show that ensemble methods combining $m$ single learners (ExtBagging,Snapshot,DynSnap-cyc,DynSnap-step) consistently outperform the methods with a single learner (SingleBase and MCDropout) for all metrics and all datasets. This demonstrates Theorem 1 that an ensemble learner can achieve better consistency than individual learners. Except MCDropout, the ACC improvements compared to SingleBase on CIFAR-10, CIFAR-100, and Yahoo!Answers are in the range 1.8%-3.3%, 5.4%-8.3%, and 0.7%-2%, respectively. CON improvements are 4.5%-8.2%, 9.8%-19.3%, and 3.4%-4.5%. ACC-CON improvements are 3.7%-6.5%, 8.3%-14.1%, and 1.4%-2.3%. ExtBagging, DynSnap-cyc and DynSnap-step perform the best, followed by Snapshot, while MCDropout fails compared to SingleBase in this experimental setting. We can observe that although the improvements are not affected by the combination method, AVG and WAVG perform slightly better than MV and WMV. Results for additional metrics are presented in Appendix K, where we can make similar observations. From Table 2, which shows the training time of all ensemble methods relative to SingleBase, we can observe that our proposed ensemble methods greatly reduce the training time compared with ExtBagging with comparable predictive performance.

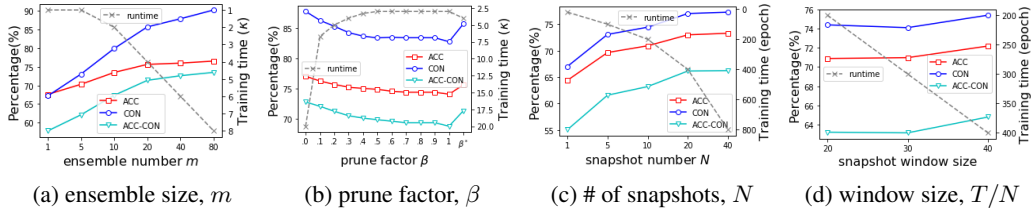

| (a) ensemble size, $m$ | (b) prune factor, $\beta$ | (c) # of snapshots, $N$ | (d) window size, $T/N$ |
|---|---|---|---|

Figure 1: Sensitivity analysis on CIFAR100+ResNet56 using DynSnap-cyc with AVG combination. (a) Varying ensemble size $m$ with $\beta = \beta^*, N = 10, T/N = 20$. (b) Varying prune factor $\beta$ with $m = 20, N = 10, T/N = 20$. (c) Varying snapshot number $N$ with $m = N, \beta = 1, T/N = 20$. (d) Varying snapshot window size $T/N$ with $m = 20, \beta = 1, N = 10$.

*Discussion*: According to our theorems, ExtBagging is expected to perform well across all metrics as it selects components with the best estimated accuracy, but is an expensive procedure. Further, random initialization of model parameters and random shuffle of training data result in data diversity and parameter diversity, which are key factors of a successful ensemble [31]. Snapshot meanwhile performs well on all metrics at no additional cost, however, it lacks data diversity. MCDropout cannot guarantee sampling of good components during inference time thus may not satisfy Corollary 5.1, and may diminish the model performance in our setting. Our proposed DynSnap-cyc and DynSnap-step combines ExtBagging and Snapshot techniques thus combining the strengths of both methods and hence results in better performance with less training cost. These observations validate our theorems empirically.

**Sensitivity analysis on hyper-parameters**: Figure 1 shows the effect of hyper-parameters - ensemble size $m$ (Figure 1a), prune factor $\beta$ (Figure 1b), number of snapshots $N$ (Figure 1c), snapshot window size $T/N$ (Figure 1d) - on the performance of CIFAR100+ResNet56 using DynSnap-cyc model with AVG combination. Experiment details are presented in Appendix M.

*Major observations and discussions*: (1) Figure 1a shows prominent improvements for all metrics as $m$ increases. This demonstrates the Theorem 1 and Theorem 2 that better consistency can be achieved if more component learners are combined. (2) Figure 1b shows that performance decreases as $\beta$ increases. A larger $\beta$ means a minor pruning after one snapshot learning, which further means adding relatively poor trained learners in the ensemble $\zeta$, and vice versa. The results validate Corollary 5.1 empirically. (3) The results in Figure 1c show a performance improvement as $N$ doubles at the cost of extra training time, however, the improvement rate declines after N=20. Figure 1d shows that the snapshot window size has small impact on the model performance. This indicates that using orginal Snapshot [19] method (i.e. DynSnap-cyc with $\beta = 1$), the improvement reaches an upper bound after a certain period. Our proposed DynSnap-cyc that combines ExtBagging and Snapshot, has further improved the performance compared to Snapshot and is more efficient than ExtBagging.

# 7 Conclusion and future work

In this work, we defined *consistency* and *correct-consistency* of a model and metrics to measure them, and provided a theoretical explanation of why and how an ensemble learner can achieve better consistency and correct-consistency than the average performance of individual learners. We proposed an efficient dynamic snapshot ensemble method that conforms with our theory and demonstrated the value of the proposed method and theorems on multiple datasets. In the future, we plan to explore the following directions: 1) Theoretically prove an upper bound on the number of learners as a function of the dataset characteristics (size, labels, etc.). 2) Incorporate consistency metrics in model training and optimize for it in addition to accuracy.

## Broader Impact

Users' trust in AI systems is of paramount importance and is the genesis of the problem discussed in this study. We developed metrics to measure consistency as well as correct-consistency of the outcome of AI systems. The problem discussed in this paper is foundational in nature, however, we firmly believe that the study may have a major impact on applications of prognostics and recommendation

engines where users' behavior or actions can be affected immediately. Examples of such applications include but are not limited to vehicle/industrial asset repair recommendations, medical diagnosis recommendations, failure predictions, vehicle safety alerts, epidemic onset/peak time forecasting, traffic predictions, etc. Our recommendation is to use the proposed metrics in addition to existing aggregate metrics like accuracy to evaluate the AI system before deployment. The Data scientist and DevOps team may consider highlighting these metrics to decision makers before deployment. While this work highlights the importance of consistency and correct-consistency, care should be taken to study and quantify generalization capability of the AI models. An AI model with 100% accuracy generally indicates possible overfitting, and hence, appropriate trade-off between consistency metrics and generalization metrics may be required. Moreover, careful weighting of these metrics should be considered in applications by decision makers. Thus, we also further recommend to emphasize on continual learning research by the AI community and incorporate our proposed metrics appropriately. Furthermore, we emphasize that the proposed work is applicable to other non-neural network algorithms such as XGBoost which does not have a closed form solution. This may be an immediate future direction to extend this study. In conclusion, we strongly recommend further development in this newly introduced impactful research direction before any real deployment to critical applications.

## Acknowledgments and Disclosure of Funding

This work was initiated when Lijing Wang was at Industrial AI Lab of Hitachi America Ltd. as a research intern and completed when she was back to University of Virginia. The authors are grateful towards HAL Research and Development for funding the research, providing computing resources and for constructive feedback during various discussions. The authors also want to thank researchers of the Network Systems Science and Advanced Computing at Biocomplexity Institute and Initiative of University of Virginia for their support and valuable feedback. Lijing Wang also was partially supported by NSF Grant No OAC-1916805, NSF Expeditions in Computing Grant CCF-1918656, CCF-1917819. Any opinions, findings, and conclusions or recommendations expressed in this material are those of the author(s) and do not necessarily reflect the views of the funding agencies and organization. The authors are also grateful to NeurIPS reviewers for their critical and constructive feedback.

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
