[Supplementary Material]

# A  List of Notations

Table 1: Notations and their meanings

| Notation | Meaning |
|---|---|
| $C = \{C_1, C_2, \ldots, C_p\}$ | Classification problem with $p$ class labels, $C_k$; $(1 \le k \le p$ and $2 \le p)$ |
| $I = \{I_1, I_2, \ldots, I_n\}$ | Testing instance, $I_t$; $(1 \le t \le n)$ |
| $r_t = \langle R_t^1, R_t^2, \ldots, R_t^p \rangle$ | Ground truth one-hot vector for $I_t$ |
| $D = \{D_1, D_2, \ldots, D_T\}$ | Stream training dataset, $D_i$; $(1 \le i \le T)$ $D_i \subseteq D_j$ if $i < j$ |
| $L = \{L_1, L_2, \ldots, L_T\}$ | Trained learners of a model for $D$, $L_i$; $(1 \le i \le T)$ |
| $\hat{Y} = \{\hat{Y}_1, \hat{Y}_2, \ldots, \hat{Y}_T\}$ | Prediction for $I$ by $L$, $\hat{Y}_i$; $(1 \le i \le T)$ |
| $\xi = \{SL_1, SL_2, \ldots, SL_m\}$ | Ensemble of $m$ component single learners, $SL_j$; $(1 \le j \le m$ and $2 \le m)$ |
| $\tilde{\xi} = \{\tilde{SL}_1, \tilde{SL}_2, \ldots, \tilde{SL}_m\}$ | The copy of $\xi$, ensemble of $m$ component single learners, $\tilde{SL}_j$; $(1 \le j \le m$ and $2 \le m)$ |
| $s_{tj} = \langle S_{tj}^1, S_{tj}^2, \ldots, S_{tj}^p \rangle$ | Prediction vector for $I_t$ by $SL_j$; $\sum_{k=1}^p S_{tj}^k = 1$ |
| $\tilde{s}_{tj} = \langle \tilde{S}_{tj}^1, \tilde{S}_{tj}^2, \ldots, \tilde{S}_{tj}^p \rangle$ | Prediction vector for $I_t$ by $\tilde{SL}_j$; $\sum_{k=1}^p \tilde{S}_{tj}^k = 1$ |
| $o_t = \langle O_t^1, O_t^2, \ldots, O_t^p \rangle$ | Centroid-point vector for $I_t$ by $\xi$, $O_t^k$; $(1 \le k \le p)$ |
| $\tilde{o}_t = \langle \tilde{O}_t^1, \tilde{O}_t^2, \ldots, \tilde{O}_t^p \rangle$ | Centroid-point vector for $I_t$ by $\tilde{\xi}$, $\tilde{O}_t^k$; $(1 \le k \le p)$ |
| $w = \langle W_1, W_2, \ldots, W_m \rangle$ | Weight vector for $\xi$, $W_j$; $(1 \le j \le m)$ |
| $\tilde{w} = \langle \tilde{W}_1, \tilde{W}_2, \ldots, \tilde{W}_m \rangle$ | Weight vector for $\tilde{\xi}$, $\tilde{W}_j$; $(1 \le j \le m)$ |
| $\zeta = f(w, \xi)$ | Combination learner for $\xi$ and $w$ |
| $\tilde{\zeta} = f(\tilde{w}, \tilde{\xi})$ | Combination learner for $\tilde{\xi}$ and $\tilde{w}$ |

# B  Proof of Theorem 1

*Proof.* Based on Minkowski's inequality for sums [2] with order 2:

$$\sqrt{\sum_{k=1}^p (\sum_{j=1}^m \theta_j^k)^2} \le \sum_{j=1}^m \sqrt{\sum_{k=1}^p (\theta_j^k)^2} \tag{8}$$

Letting $\theta_j^k = S_{tj}^k - \tilde{S}_{tj}^k$ and substituting in Eq. 8

$$\sqrt{\sum_{k=1}^p (\sum_{j=1}^m (S_{tj}^k - \tilde{S}_{tj}^k))^2} \le \sum_{j=1}^m \sqrt{\sum_{k=1}^p (S_{tj}^k - \tilde{S}_{tj}^k)^2} \tag{9}$$

Since $m > 0$, we have the following

$$\sqrt{\sum_{k=1}^p (m \frac{1}{m} \sum_{j=1}^m (S_{tj}^k - \tilde{S}_{tj}^k))^2} \le \sum_{j=1}^m \sqrt{\sum_{k=1}^p (S_{tj}^k - \tilde{S}_{tj}^k)^2}$$

$$\Rightarrow m \sqrt{\sum_{k=1}^p (\frac{1}{m} \sum_{j=1}^m (S_{tj}^k - \tilde{S}_{tj}^k))^2} \le \sum_{j=1}^m \sqrt{\sum_{k=1}^p (S_{tj}^k - \tilde{S}_{tj}^k)^2}$$

$$\Rightarrow \sqrt{\sum_{k=1}^p (\frac{1}{m} \sum_{j=1}^m S_{tj}^k - \frac{1}{m} \sum_{j=1}^m \tilde{S}_{tj}^k)^2} \le \frac{1}{m} \sum_{j=1}^m \sqrt{\sum_{k=1}^p (S_{tj}^k - \tilde{S}_{tj}^k)^2}$$

Using Eq. 1 and 3, Eq. 4 can be proved. □

# C  Proof of Theorem 2

*Proof.* Since $o_t$ is the centroid-vector of all $o_{tl}$ vectors, assume $\xi_l$ as an individual component learner with prediction vector of $o_{tl}$, Eq. 5 holds for every $\xi_l$ according to Theorem 1. □

# D  Proof of Theorem 3 and 4

Consider the error of a prediction for a given instance $t$ be represented as the Euclidean distance between the prediction vector $s_{tj}$ and ground truth vector $r_t$, denoted as $distance(s_{tj}, r_t) = $

$\sqrt{\sum_{k=1}^{p}(S_{tj}^{k} - R_{t}^{k})^2}$, Bonab and Can [7] prove that

$$distance(o_t, r_t) \leq \frac{1}{m} \sum_{j=1}^{m} distance(s_{tj}, r_t) \tag{10}$$

where a smaller distance means a smaller error. Similarly, for the prediction vector $\tilde{s}_{tj}$, we have $distance(\tilde{o}_t, r_t) \leq \frac{1}{m} \sum_{j=1}^{m} distance(\tilde{s}_{tj}, r_t)$. Using Eq. 3, 10, and 2, we have the following

$$distance(o_t, \tilde{o}_t) + distance(o_t, r_t) + distance(\tilde{o}_t, r_t)$$

$$\leq \frac{1}{m} \sum_{j=1}^{m} distance(s_{tj}, \tilde{s}_{tj}) + \frac{1}{m} \sum_{j=1}^{m} distance(s_{tj}, r_t) + \frac{1}{m} \sum_{j=1}^{m} distance(\tilde{s}_{tj}, r_t)$$

$$\Rightarrow distance(o_t, \tilde{o}_t, r_t) \leq \frac{1}{m} \sum_{j=1}^{m} distance(s_{tj}, \tilde{s}_{tj}, r_t)$$

Eq. 6 can be proved. Similar to proof in C, Theorem 4 can be proved.

## E    Generalization of Theorem 1, 2 and Theorem 3, 4

Theorem 1, 2 and Theorem 3, 4 can be generalized to Minkowski distance with order $q, q > 1$. We use Minkowski distance to represent the distance between $s_{tj}$ and $\tilde{s}_{tj}$, which is denoted as:

$$distance(s_{tj}, \tilde{s}_{tj}) = \left( \sum_{k=1}^{p} |S_{tj}^{k} - \tilde{S}_{tj}^{k}|^q \right)^{\frac{1}{q}} \tag{11}$$

where $q = 2$ is Euclidean distance.

*Proof.* By replacing order 2 with order $q$ in B, we have

$$\left( \sum_{k=1}^{p} |\frac{1}{m} \sum_{j=1}^{m} S_{tj}^{k} - \frac{1}{m} \sum_{j=1}^{m} \tilde{S}_{tj}^{k}|^q \right)^{\frac{1}{q}} \leq \frac{1}{m} \sum_{j=1}^{m} \left( \sum_{k=1}^{p} |S_{tj}^{k} - \tilde{S}_{tj}^{k}|^q \right)^{\frac{1}{q}} \tag{12}$$

Using Eq. 11 and 3, Eq. 4 can be proved. Thus Theorem 1 still holds in this case, as well as Theorem 2 and Theorem 3, 4. $\qquad \square$

## F    Proof of Theorem 5

*Proof.* Let $A$ and $B$ be the subsets of $I$ that are correctly predicted by $\zeta$ and $\tilde{\zeta}$. Then $acc_\zeta = \frac{|A|}{n}$, $acc_{\tilde{\zeta}} = \frac{|B|}{n}$, and $ccon(\zeta, \tilde{\zeta}) = \frac{|A \cap B|}{n}$. Since $|A| + |B| - |A \cap B| = |A \cup B| \leq n$, we have $\frac{|A|}{n} + \frac{|B|}{n} - 1 \leq \frac{|A \cap B|}{n}$, i.e. $(acc_\zeta + acc_{\tilde{\zeta}}) - 1 \leq ccon(\zeta, \tilde{\zeta})$. And we always have $0 \leq ccon(\zeta, \tilde{\zeta})$. So we prove the left inequality. Now notice that $|A \cap B| \leq |A|$ and $|A \cap B| \leq |B|$, we also prove the right inequality. $\qquad \square$

## G    Proof of Corollary 5.1

To be consistent with accuracy definition, we denote the correctness of $s_{tj}$ for instance $t$ as $sim(s_{tj}, r_t) = (\sqrt{2} - distance(s_{tj}, r_t))/\sqrt{2}$ where $sim(s_{tj}, rt)$ is in the range $[0, 1]$ and $distance(s_{tj}, r_t)$ is in range $[0, \sqrt{2}]$, $\sqrt{2}$ is the largest Euclidean distance in the probability simplex. Given a test dataset $I$, the correctness of a learner $SL_j$ on $I$ can be denoted as

$corr_{SL_j} = \frac{1}{n}\sum_{t=1}^{n} sim(s_{tj}, r_t)$. Based on 10, we have the following:

$$\frac{1}{n}\sum_{t=1}^{n}\frac{1}{m}\sum_{j=1}^{m} sim(s_{tj}, r_t) \leq \frac{1}{n}\sum_{t=1}^{n} sim(o_t, r_t)$$

$$\Rightarrow \frac{1}{m}\sum_{j=1}^{m}(\frac{1}{n}\sum_{t=1}^{n} sim(s_{tj}, r_t)) \leq \frac{1}{n}\sum_{t=1}^{n} sim(o_t, r_t)$$

Thus,

$$\frac{1}{m}\sum_{j=1}^{m} corr_{SL_j} \leq corr_\zeta \tag{13}$$

where $(0 \leq corr \leq 1)$ and a larger $corr$ means a better correctness. Here, $s_{tj}$ and $o_t$ are $p$-dimension vectors which could be a one-hot vector or not.

According to accuracy definition in our paper, $acc_{SL_j} = \frac{1}{n}\sum_{t=1}^{n} 1^1_{SL_j,r}(t)$, $acc_\zeta = \frac{1}{n}\sum_{t=1}^{n} 1^1_{\zeta,r}(t)$ where $1^1_{SL_j,r}(t)$ and $1^1_{\zeta,r}(t)$ are defined in 17. There is a discrepancy between $corr_\zeta$ and $acc_\zeta$ as the latter one converts the predicted vector $o_t$ to one-hot vector by using $argmax(o_t)$. Thus, Eq. 13 is *not* equivalent to

$$\frac{1}{m}\sum_{j=1}^{m} acc_{SL_j} \leq acc_\zeta \tag{14}$$

However, assuming $s_{tj}$ is a one-hot vector, then $\frac{1}{m}\sum_{j=1}^{m} corr_{SL_j} = \frac{1}{m}\sum_{j=1}^{m} acc_{SL_J}$. If we can prove that $corr_\zeta \leq acc_\zeta$ is true with some conditions, then Eq. 14 is true.

Let $argmax(r_t) = g_t$, $g_t \in \{1, \ldots, p\}$ denote the label of instance $I_t$, given $\zeta$ and $I_t$, the probability $P(y = g_t|\zeta, I_t) = O_t^{g_t}$. For $I$, (*i*) if for $\forall I_t \in I$ that $argmax(o_t) = g_t$ is true (the probability is $\prod_{t=1}^{n} O_t^{g_t}$), then $corr_\zeta \leq acc_\zeta = 1$ is true with probability $\eta = \prod_{t=1}^{n} O_t^{g_t}$, $0 \leq \eta \leq 1$; (*ii*) else if for $\forall I_t \in I$ that $argmax(o_t) \neq g_t$ is true (the probability is $\prod_{t=1}^{n}(1 - O_t^{g_t})$), then $0 = acc_\zeta \leq corr_\zeta$ is true with probability $\bar{\eta} = \prod_{t=1}^{n}(1 - O_t^{g_t})$, $0 \leq \bar{\eta} \leq 1$; (*iii*) otherwise, if $\exists I_t \in I$ that $argmax(o_t) = g_t$ is true (the probability is $1 - \eta - \bar{\eta}$), then $corr_\zeta \leq acc_\zeta$ is true with probability $\epsilon$ $(0 \leq \epsilon \leq 1 - \eta - \bar{\eta})$ which is not quantified in this work. According to (*i*) and (*iii*), we can say that $corr_\zeta \leq acc_\zeta$ is true with probability $\eta + \epsilon$, denoted as $\iota$.

**Claim 3.1.1.** Assuming $s_{tj}$ $(t \in \{1, \ldots, n\}, j \in \{1, \ldots, m\})$ is a one-hot vector, Eq. 14 is true with probability $\iota$, and **at least** with probability $\eta$. **Note that $\iota$ can empirically be estimated as $acc_\zeta$.**

The above proof shows that a better accuracy of an ensemble can be achieved by combining components with accuracy that is at least equal to the average accuracy of individual components (i.e. increasing the lower bound of 14). Based on this, let $a = \frac{1}{m-1}\sum_{SL_i \in \xi_l} acc_{SL_i}$, $a^+ = \frac{1}{m}\sum_{j=1}^{m} acc_{SL_j}$, then according to Eq. 14, we have $a \leq acc_{\zeta_l} \leq 1$ and $a^+ \leq acc_\zeta \leq 1$.

**Claim 3.1.2.** Assuming that $acc_{\zeta_l}$ and $acc_\zeta$ are uniformly distributed, if $a \leq acc_{SL_l}$, then $a \leq a^+$. Then we have $acc_{\zeta_l} \leq acc_\zeta$ with probability $\varepsilon = \frac{a^+ - a}{1-a} + \frac{1}{2} * \frac{1-a^+}{1-a}$.

Claim 3.1.1 and 3.1.2 also hold for $acc_{\tilde{\zeta}}$ and $acc_{\tilde{\zeta}_l}$ with probability $\tilde{\eta}$ and $\tilde{\varepsilon}$ calculated using corresponding items. We omit the calculation for brevity.

According to Theorem 5, let $b^+ = max(acc_\zeta + acc_{\tilde{\zeta}} - 1, 0)$, $c^+ = min(acc_\zeta, acc_{\tilde{\zeta}})$, $b = max(acc_{\zeta_l} + acc_{\tilde{\zeta}_l} - 1, 0)$, and $c = min(acc_{\zeta_l}, acc_{\tilde{\zeta}_l})$, then we have

$$b \leq ccon(\zeta_l, \tilde{\zeta}_l) \leq c$$

and

$$b^+ \leq ccon(\zeta, \tilde{\zeta}) \leq c^+$$

If $acc_{\zeta_l} \leq acc_\zeta$, then $b \leq b^+$ and $c \leq c^+$. Assuming that $ccon(\zeta, \tilde{\zeta})$ and $ccon(\zeta_l, \tilde{\zeta}_l)$ are uniformly distributed, we have the following:

(1) If $c \leq b^+$, then $ccon(\zeta_l, \tilde{\zeta}_l) \leq ccon(\zeta, \tilde{\zeta})$ with probability $\frac{1}{2}$;

(2) Otherwise, if $ccon(\zeta, \tilde{\zeta})$ is between $[b, b^+]$ or $ccon(\zeta_l, \tilde{\zeta}_l)$ is between $[c, c^+]$, then $ccon(\zeta_l, \tilde{\zeta}_l) \leq ccon(\zeta, \tilde{\zeta})$ with probability $\frac{1}{2} * (\frac{b^+ - b}{c - b} + \frac{c^+ - c}{c^+ - b^+})$; or if $ccon(\zeta, \tilde{\zeta})$ is between $[b^+, c]$ and $ccon(\zeta_l, \tilde{\zeta}_l)$ is between $[b^+, c]$, then $ccon(\zeta_l, \tilde{\zeta}_l) \leq ccon(\zeta, \tilde{\zeta})$ with probability $\frac{1}{2} * \frac{1}{2} * (\frac{c - b^+}{c - b} * \frac{c - b^+}{c^+ - b^+})$.

**Claim 3.1.3.** If $acc_{\zeta_l} \leq acc_\zeta$, then $ccon(\zeta_l, \tilde{\zeta}_l) < ccon(\zeta, \tilde{\zeta})$ with probability $\upsilon = \frac{1}{2} + \frac{1}{2} * (\frac{b^+ - b}{c - b} + \frac{c^+ - c}{c^+ - b^+}) + \frac{1}{2} * \frac{1}{2} * (\frac{c - b^+}{c - b} * \frac{c - b^+}{c^+ - b^+})$.

According to Claim 3.1.1 and 3.1.2 and 3.1.3, we prove Corollary 5.1 at least with the probability $\rho = \eta \tilde{\eta} \varepsilon \tilde{\varepsilon} \upsilon$. Note that $\rho$ provides a lower bound of the probability that Corollary 5.1 is true.

# H  Metrics

In this section, we define multiple metrics for consistency, accuracy, and correct-consistency in detail. Consider two learners $A$ and $B$, the prediction vectors of $A$,$B$ for a data point $I_t$ are denoted as $y_t^A$ and $y_t^B$. Here $A$,$B$ could be any single learner or ensemble learner, and $y_t^A$, $y_t^B$ could be a single prediction or a combined prediction for $I_t$ with true label $r_t$. We define indicator functions $1_{A,B,r}(\cdot), 1_{A,B}^k(\cdot)$, and $1_{A,r}^k(\cdot)$ as:

$$1_{A,B,r}(t) = \begin{cases} 1, & \text{if } argmax(y_t^A) = argmax(y_t^B) = argmax(r_t) \\ 0, & otherwise \end{cases} \quad (15)$$

where $argmax(\cdot)$ returns the index of max value in a list, which indicates the class label.

$$1_{A,B}^k(t) = \begin{cases} 1, & \text{if } \exists\, 0 \leq i, j \leq k \text{ that } argmax_i(y_t^A) = argmax_j(y_t^B) \\ 0, & otherwise \end{cases} \quad (16)$$

$$1_{A,r}^k(t) = \begin{cases} 1, & \text{if } \exists\, 0 \leq i \leq k \text{ that } argmax_i(y_t^A) = argmax(r_t) \\ 0, & otherwise \end{cases} \quad (17)$$

where $argmax_i(\cdot)$ returns the index of the $i$-th max value in a list.

Based on the above definitions, for a testing data set $I$, the metrics used to evaluate consistency, accuracy, correct-consistency are computed as:

- Consistency (CON):

$$\frac{1}{n} \sum_{t=1}^{n} 1_{A,B}^1(t) \quad (18)$$

- Accuracy (ACC):

$$\frac{1}{n} \sum_{t=1}^{n} 1_{A,r}^1(t) \quad (19)$$

- Correct-Consistency (ACC-CON):

$$\frac{1}{n} \sum_{t=1}^{n} 1_{A,B,r}(t) \quad (20)$$

- Coarse-consistency (CCON-K), also called TopK-consistency:

$$\frac{1}{n} \sum_{t=1}^{n} 1_{A,B}^k(t) \quad (21)$$

- Coarse-accuracy (CACC-K), also called TopK-accuracy:

$$\frac{1}{n} \sum_{t=1}^{n} 1_{A,r}^k(t) \quad (22)$$

- Pearson's r coefficient (Pearson). Computing the similarity between two vectors.

$$\frac{1}{n}\sum_{t=1}^{n}\frac{\sum(y_t^A - \bar{y}_t^A)(y_t^B - \bar{y}_t^B)}{\sqrt{\sum(y_t^A - \bar{y}_t^A)^2}\sqrt{\sum(y_t^B - \bar{y}_t^B)^2}} \qquad (23)$$

where $\bar{y}_t^A$ is the average value of all elements in $y_t^A$.

- Cosine similarity (Cosine). Computing the cosine similarity between two vectors.

$$\frac{1}{n}\sum_{t=1}^{n}\frac{\sum y_t^A y_t^B}{\sqrt{\sum(y_t^A)^2}\sqrt{\sum(y_t^B)^2}} \qquad (24)$$

where $\sum y_t^A y_t^B$ denotes the summation of the element-wise products.

# I Experiment design

Fig. 2 shows the metrics computation in our experiments. The accuracy (ACC) reported in this paper is the average of three individual accuracy on $D_1$, $D_2$, $D_3$, while the consistency (CON) and correct-consistency (ACC-CON) are the average of three values computed from any two of $D_1$, $D_2$, $D_3$. We have created a git repository for this work and will be posted upon the acceptance and publication of this work.

Figure 1: Metrics computation in the experiments.

# J Dataset Design

To simulate the online training environment, given a dataset we generate three class imbalanced training datasets $D_1, D_2, D_3$, where $D_1 \subseteq D_2 \subseteq D_3$. The testing set $I$ is independent of training data and covers the minimum number of classes in any of the three training sets. The validation set is sampled from $D_i$ and has the same sample size and class distribution as $I$. Each class imbalanced training dataset is generated by randomly varying the number of samples for each class. For each class, a percentage is manually specified to create the imbalanced class distribution. Take CIFAR10 as an example, we specify a percentage value for each class, i.e. $P = \{1, 0.9, 0.8, 0.95, 0.45, 0.3, 0.4, 0.1, 0.85, 0.75\}$. The original training dataset is used to generate three sub-datasets - $T_1, T_2, T_3$, where $T_1$ has 9 classes (horse excluded in our experiments) and $6000 * P_i * 0.8$ images per class, $T_2$ has 9 classes and $6000 * P_i * 1.0$ images per class, $T_3$ has 10 classes and $6000 * P_i * 1.0$ images per class. It should be noted that the validation and testing datasets are from the original testing dataset before extraction of the above-mentioned training data sub-sets. The validation and testing datasets have 9 classes and 500 images per class. Training dataset $T_1$ and validation dataset together constitute $D_1$. During model training procedure, we sample validation dataset from $D_1$ without changing the class distribution. Similar processes for $D_2$ and $D_3$ are followed. All the scripts to reproduce the imbalanced datasets and the proposed method in our experiments will be posted upon the acceptance and publication of this work.

## K    Additional metric results

In this section, we present additional results for metrics defined in H in Table 4, i.e. CCON-2, Pearson, Cosine, CACC-2. Only AVG and WAVG are evaluated because MV and WMV are one-hot encoding predicted vectors. From the results we can derive the same conclusion as described in Section 6.

Table 2: Additional consistency and accuracy results.

| | CIFAR10 ResNet20 | | CIFAR100 ResNet56 | | YAHOO!Answers fastText | |
|---|---|---|---|---|---|---|
| **CCON-2**(%) | AVG | WAVG | AVG | WAVG | AVG | WAVG |
| SingleBase | 98.55 | 98.55 | 88.30 | 88.30 | 99.45 | 99.45 |
| ExtBagging | 99.68 | 99.68 | 98.11 | 98.11 | 99.80 | 99.80 |
| MCDropout | 98.52 | 98.52 | 87.72 | 87.72 | 98.94 | 98.94 |
| Snapshot | 99.31 | 99.33 | 93.93 | 93.88 | 99.81 | 99.81 |
| DynSnap-cyc | 99.81 | 99.81 | 97.92 | 97.93 | 99.83 | 99.83 |
| DynSnap-step | 99.71 | 99.71 | 96.63 | 96.63 | 99.83 | 99.83 |
| **Pearson** | AVG | WAVG | AVG | WAVG | AVG | WAVG |
| SingleBase | 0.89 | 0.89 | 0.75 | 0.75 | 0.96 | 0.96 |
| ExtBagging | 0.97 | 0.97 | 0.94 | 0.94 | 0.99 | 0.99 |
| MCDropout | 0.89 | 0.89 | 0.77 | 0.77 | 0.94 | 0.94 |
| Snapshot | 0.94 | 0.94 | 0.86 | 0.86 | 0.97 | 0.97 |
| DynSnap-cyc | 0.97 | 0.97 | 0.94 | 0.94 | 0.97 | 0.97 |
| DynSnap-step | 0.96 | 0.96 | 0.91 | 0.91 | 0.98 | 0.98 |
| **Cosine** | AVG | WAVG | AVG | WAVG | AVG | WAVG |
| SingleBase | 0.89 | 0.89 | 0.75 | 0.75 | 0.96 | 0.96 |
| ExtBagging | 0.97 | 0.97 | 0.94 | 0.94 | 0.99 | 0.99 |
| MCDropout | 0.89 | 0.89 | 0.77 | 0.77 | 0.94 | 0.94 |
| Snapshot | 0.94 | 0.94 | 0.86 | 0.86 | 0.97 | 0.97 |
| DynSnap-cyc | 0.97 | 0.97 | 0.94 | 0.94 | 0.97 | 0.97 |
| DynSnap-step | 0.97 | 0.97 | 0.92 | 0.92 | 0.98 | 0.98 |
| **CACC-2**(%) | AVG | WAVG | AVG | WAVG | AVG | WAVG |
| SingleBase | 94.62 | 94.62 | 79.61 | 79.61 | 75.89 | 75.89 |
| ExtBagging | 96.59 | 96.59 | 85.74 | 85.74 | 78.16 | 78.16 |
| MCDropout | 94.48 | 94.48 | 78.96 | 78.96 | 76.69 | 76.69 |
| Snapshot | 95.53 | 95.53 | 83.45 | 83.47 | 76.93 | 76.93 |
| DynSnap-cyc | 96.30 | 96.31 | 85.61 | 85.60 | 77.44 | 77.44 |
| DynSnap-step | 96.23 | 96.23 | 84.94 | 84.94 | 77.37 | 77.37 |

## L    Additional results

Although the metrics introduced above consider the percentage of agreement in predictions and correct predictions between two models, they do not reflect upon the accuracy and percentage of correct to incorrect and incorrect to correct predictions. In general, we want more incorrect to correct (ItoC) and correct to correct (CtoC) but less correct to incorrect (CtoI) scenarios, hence we compute Com=CtoC+ItoC-CtoI as an additional metric. We show these additional results in Table 5. The observations are consistent with findings in the main paper.

Table 3: (WAVG) Percentage of CtoI, ItoC and Com predictions for $D_1 \rightarrow D_2$, $D_2 \rightarrow D_3$ and $D_1 \rightarrow D_3$.

| | $D_1 \rightarrow D_2$ | | | $D_2 \rightarrow D_3$ | | | $D_1 \rightarrow D_3$ | | |
|---|---|---|---|---|---|---|---|---|---|
| | ItoC | CtoI | Com | ItoC | CtoI | Com | ItoC | CtoI | Com |
| | **CIFAR10+ResNet20** | | | | | | | | |
| SingleBase | 6.60 | 5.38 | 80.79 | 7.08 | 6.83 | 79.59 | 5.74 | 4.27 | 82.15 |
| ExtBagging | 3.11 | 2.13 | **87.13** | 4.40 | 3.80 | **86.07** | 3.13 | 1.56 | **88.31** |
| MCDropout | 6.69 | 5.20 | 80.67 | 7.29 | 7.16 | 78.84 | 5.78 | 4.16 | 81.84 |
| Snapshot | 4.89 | 3.11 | 84.98 | 5.89 | 5.38 | 83.22 | 3.91 | 1.62 | 86.98 |
| DynSnap-cyc | 2.84 | 2.11 | 86.36 | 4.47 | 3.71 | 85.51 | 2.78 | 1.29 | 87.93 |
| DynSnap-step | 3.09 | 2.47 | 86.38 | 4.36 | 3.64 | 85.91 | 3.47 | 2.13 | 87.42 |
| | **CIFAR100+ResNet56** | | | | | | | | |
| SingleBase | 11.18 | 8.80 | 59.56 | 9.36 | 9.26 | 59.19 | 11.32 | 8.86 | 59.60 |
| ExtBagging | 6.57 | 4.42 | 70.65 | 4.87 | 4.71 | 70.53 | 6.81 | 4.51 | 70.73 |
| MCDropout | 10.02 | 9.35 | 58.08 | 10.00 | 9.21 | 59.01 | 10.69 | 9.23 | 58.99 |
| Snapshot | 8.46 | 5.92 | 67.66 | 6.32 | 5.64 | 68.63 | 8.91 | 5.68 | 68.59 |
| DynSnap-cyc | 5.74 | 3.35 | **73.19** | 3.58 | 3.90 | **72.32** | 5.60 | 3.54 | **72.69** |
| DynSnap-step | 6.44 | 5.92 | 68.14 | 5.62 | 4.93 | 69.82 | 5.98 | 4.77 | 69.98 |
| | **YAHOO!Answers+fastText** | | | | | | | | |
| SingleBase | 2.32 | 2.15 | 60.83 | 5.07 | 2.75 | 62.56 | 5.14 | 2.65 | 62.65 |
| ExtBagging | 1.72 | 1.82 | 61.54 | 4.43 | 2.29 | 63.21 | 4.65 | 2.61 | 62.88 |
| MCDropout | 2.84 | 1.65 | 62.00 | 7.88 | 5.25 | 61.04 | 7.70 | 3.88 | 62.41 |
| Snapshot | 2.05 | 1.78 | 62.07 | 3.35 | 1.56 | 64.07 | 4.48 | 2.42 | 63.21 |
| DynSnap-cyc | 1.38 | 1.05 | **63.47** | 3.46 | 1.67 | **64.63** | 4.08 | 1.97 | **64.34** |
| DynSnap-step | 1.77 | 1.28 | 63.23 | 3.52 | 1.71 | 64.62 | 4.29 | 1.99 | **64.34** |

## M  Sensitivity analysis settings

**Ensemble size** $m$**.** We set $m$ as [1, 5, 10 20, 40, 80] with $\beta = \beta^*, N = 10, T/N = 20$. **Prune factor** $\beta$**.** The prune factor $\beta$ varies from 0.0 to 1.0 with $m = 20, N = 10, T/N = 20$. **Number of snapshots** $N$**.** We vary snapshot number $N$ as $[1, 5, 10, 20, 40]$ while set $m = N, \beta = 1, T/N = 20$. We keep the window size as 20, and expand the training epochs to get more snapshot trained learners. **Snapshot window size** $T/N$**.** We set $N = 10, m = 20, \beta = 1$ but vary snapshot window size $T/N$ by varying epoch number $T$.

## N  Ablation study

We conduct ablation study on the effect of (1) learning rate schedule and (2) random shuffle of training datasets and model parameter initialization. Figure 3 show the performance on CIFAR100+ResNet56 with AVG combination. The *no-lrschedule* method is without learning rate schedule and considers top-N snapshot with rest of the model settings same as DynSnap-step. For *no-random* method, we set $m = 20, N = 40, T/N = 20$, and use top-N snapshot. From the results we observe that random shuffle of training and validation datasets has great impact on model performance. The method with random shuffle has better performance than *no-random* on three metrics. The results also show that cyclic cosine schedule is beneficial for model performance across the three metrics.

(a) ACC  (b) CON  (c) ACC-CON

Figure 2: Ablation study on the effect of (1) learning rate schedule and (2) random shuffle of training datasets and model parameter initialization using CIFAR100+ResNet56 with AVG combination.