[Reviews · NeurIPS 2020]

Review 1

Summary and Contributions: This paper introduces the concept of consistency of deep learning models and proposes a dynamic snapshot ensemble method to improve consistency. Experimental results are provided for sensitivity analysis of ensemble methods with the proposed consistency measure.

Strengths: A new formally defined consistency measure and a new ensemble method called dynamic snapshot ensemble are introduced.

Weaknesses: The major weakness is that Theorem 1, and the other theoretical results, actually show that the distance between the centroid-vectors (related to the consistency of the average predictor) is not greater than the average distance between pairs of prediction vectors. It does not imply that the consistency of the average predictor is better than any individual predictor. Hence this theoretical justification for better consistency of ensemble methods is not sufficient.

Correctness: The proofs sound correct. However, the theoretical results (e.g. theorems and corollaries) are not enough to justify the claim that ensemble methods improve consistency.

Clarity: This paper is well presented.

Relation to Prior Work: Related works are discussed.

Reproducibility: Yes

Additional Feedback:


Review 2

Summary and Contributions: -The paper proposes consistency and correct-consistency as two metrics to evaluate trustworthiness of a classification model. - Presents proofs to show that an ensemble is more consistent than individual base learners, and adding a base learner with accuracy higher than the average accuracy of the existing base learners in an ensemble results in improved correct-consistency with some probability. - Proposes a dynamic snapshot ensemble with pruning and empirically shows that its performance in terms of accuracy and correct-consistency is better than snapshot ensemble and is comparable with deep ensembles of Lakshminayaranan et. al. while being computationally more efficient.

Strengths: The paper considers the notions of consistency in terms of producing the same predictions for the same test data points when retraining a model multiple times on one training set or on a different training set. Similarly, correct consistency is defined for reproducing the correct predictions when the model is retrained. These metrics can indeed be important especially in setups where more training data becomes available for retraining periodically (see also weaknesses). The definitions of the metrics are intuitive, and the proposed algorithms are motivated by the theorems that are proved for the Euclidean distance metric for difference in predictions (probabilities). The experiments empirically validate the claims (up to some extent) and compare the proposed algorithm with three other approaches to form ensembles.

Weaknesses: Based on the defined metrics and provided theorems, the scope seems limited to single-label classification problems. Moreover, as also mentioned by the authors in the paper, an important application of this notion of consistency is when more training data becomes available. Although the introduced metrics consider the percentage of agreement in predictions/correct predictions between two models, they do not necessarily encourage reproducing all correct predictions in addition to giving more new correct predictions. The experiments need to show how the accuracy and percentage of correct->incorrect and incorrect->correct predictions change for D1->D2, and D2->D3 using different approaches to form ensembles. Additionally, the effect of the random initialization and shuffling in the algorithm is not clear based on the experiments. How would the results look if a larger N is used and the top half best performing learners are kept for the ensemble?

Correctness: The claims and method, and the empirical evaluations seem sound and correct. The experiments results can be improved for better illustration of the comparisons between methods as mentioned above.

Clarity: The paper is written pretty clearly. Some experimental details for reproducibility are missing, although the authors have said the code for the experiments will be provided which should solve this issue.

Relation to Prior Work: The paper discusses related work and differentiates itself from former notions of reproducibility and discusses similarities and differences from previous ensemble methods in DL.

Reproducibility: No

Additional Feedback: Update after rebuttal: I read the authors’ response, but I still have my main concerns. As mentioned in my review, the introduced metrics consider the percentage of agreement in predictions/correct predictions between two models, but they do not necessarily encourage reproducing all correct predictions in addition to giving more new correct predictions. The provided results in the authors’ rebuttal in fact seem to show this. Although the proposed method has higher correct->correct percentages, the percentages of incorrect->correct predictions for the proposed method are among the lowest. It is desirable to also have higher percentages of incorrect->correct as more training data becomes available, which is an important application as also mentioned by the authors in the paper (like the paper’s experiments setup where D1 is a subset of D2 and D2 is a subset of D3). Moreover, still the effect of the random initialization and shuffling in the proposed algorithm is not clear based on the experiments. An ablation study can be setting N=2m and keeping the top performing half (m). The current ablation studies consider snapshot without any pruning, i.e. larger N with beta=1 (keeping all).


Review 3

Summary and Contributions: Additional comments: - The terms "significant improvements (L282)", "significantly reduce (L298)", and "significant performance improvement (L303)" should be used very carefully because such terms can mislead readers because they may think the results are STATISTICALLY significant. - "ACC and CON seem to be correlated with each other" -- The first example provided by the author is not a good example as their differences are very subtle. The second example seems to be unrelated with my concern. - Other concerns are well addressed. In summary, I have a concern about the term author used in the paper. Please consider using a different term. A concern has not been addressed but other concerns have been addressed. My overall score is not changed. ---------- The authors point out the importance of producing consistently correct outputs for improving a model’s trustworthiness. And then they theoretically and empirically investigate why and how ensembles improve consistency and correct-consistency of DL. The authors formally define the consistency of a DL classifier and provide theoretical and empirical explanations about improving the consistency of DL classifiers, which can bring more attention to consistency estimation in DL.

Strengths: - This paper formally defines the consistency of a classifier and investigates the benefits of ensemble approaches for improving consistency. The concept of consistency has been discussed, but not many works have systemically defined it and performed a theoretical and empirical assessment of it. This is an interesting and timely appropriate problem for improving the trustworthiness of DL. - The proposed method is intuitive and built on clear theoretical motivation.

Weaknesses: - The proposed method is not significantly powerful than the ExtBagging in terms of accuracy and consistency. - In table 2, ACC and CON seem to be correlated with each other. Comparison with methods having similar accuracy with DynSnap but significantly lower consistency than DynSnap will be more interesting and help to support the usefulness of the consistency measure. Based on the current experimental results itself, CON seems to be redundant in evaluating ensemble approaches because ACC and CON seem to be correlated.

Correctness: Yes.

Clarity: Yes. This paper is very well written. - New terms are clearly defined. - The scope of the paper is well clarified.

Relation to Prior Work: Yes.

Reproducibility: Yes

Additional Feedback: - Generalization of consistency: Do the theoretical findings work correctly when any other distance measure is used to define consistency? Minor: (1) Table 2: Bold the highest hits (2) Provide more details about the replication results. How many replications did the author perform?


Review 4

Summary and Contributions: The paper formally defines and studies ‘consistency’ and ‘correct-consistency’ in the context of periodic retraining of deployed model where the outputs from successive generation might not agree on the correct labels assigned to the same input. Authors presents an ensemble-based technique to improve ‘consistency’ and ‘correct-consistency’, provide a theoretical explanation and validate the proposed approach on several datasets.

Strengths: The paper formally defines and tackles the ‘consistency’ and ‘correct-consistency’ of a model over successive generation. It is an important problem to solve since consistent model behavior is important to increase user’s trust. These terms are well defined with appropriate examples. Author gives a theoretical justification of why ensemble-based techniques increase ‘consistency’ and ‘correct-consistency’ and validate these theorems by extensive experiments on three datasets. Based on the theoretical justification authors propose a novel ensemble-based technique to improve ‘correct-consistency’ and compared to a bagging approach in which ensemble models are trained from the scratch with random initialization and shuffle of the data. The proposed approach is able to achieve comparable correct-consistency with much lesser training time.

Weaknesses: Motivation behind some parts of the proposed approach is not clearly mentioned. For example, it is not mentioned why having learning rate scheduling was important. Also, having an ablation to understand contribution of these component might help in understanding the proposed approach better. Authors did not discuss how does the theoretical justification work in a setting where training data distribution changes over successive model generations. In empirical evaluation they have taken this factor into the account though.

Correctness: The claims and empirical methodology used in the paper are correct.

Clarity: The paper is well written and the proposed technique and experimentations are easy to understand.

Relation to Prior Work: There is not much prior work in the direction of ‘correct-consistency’ but authors do mention related prior work related to reproducibility and ensemble-based approaches in deep learning.

Reproducibility: Yes

Additional Feedback:

[Author Response · NeurIPS 2020]

The authors would like to thank the reviewers for their constructive feedback. Our response (for each reviewer) follows.

**Reviewer #1** points out that "the proposed theoretical results do not imply that the consistency of ensemble methods are better than any individual predictor". We agree and we will refine the explanation further and state that the method is appropriate when average performance of all the predictors in the ensemble is considered. However, it should be noted that this is a well-known issue when using ensemble methods to improve other performance metrics as well (e.g. accuracy) and has been discussed in previous research. In 'R.Polikar,*Ensem. Based Syst. in Decis. Makin.*,IEEE Circ. and Syst. Magaz.,6(3):21-45,2006', 'R. Polikar, *Ensem. Learn.*, Scholarpedia,4(1):2776,2009', authors state that *"...there is no guarantee that the combination of multiple classifiers will always perform better than the best individual classifier in the ensemble. Nor an improvement on the ensemble's average performance can be guaranteed except for certain special cases. ...it certainly reduces the overall risk of making a particularly poor selection."* Our theoretical results are consistent with the above statements and show ensemble methods can but not necessarily be better than the best component. In practice for online systems during model re-training we cannot know the best possible model in advance, hence for practical reasons we should compare with average performance of all predictors in the ensemble. Further, we provide a theoretical support (Corollary 3.1) to show how an improvement on the ensemble's performance (correct-consistency) can be guaranteed with a quantifiable probability. We will clarify this in the revised version and also add the statement 'combining classifiers may not necessarily beat the consistency performance metric of the best classifier in the ensemble'. We hope the introduction of an important problem, originality and strong empirical feasibility of our work will spark more interest in consistency estimation.

**Reviewer #2** points out that "the scope of our metrics and theorems seem limited to single-label classification problems". We agree and stated in lines 145-147 (page 4), multi-label classification is a challenging problem where the statement *"A smaller Euclidean distance corresponds to a higher consistency."* is not always true. The problem will be explored in the future. Reviewer suggested to show "how the accuracy and percentage of correct->incorrect and incorrect->correct predictions change for D1->D2 and D2->D3". We show these additional results in Table 1. In general, we want more incorrect->correct (ItoC) and correct->correct (CtoC) but less correct-incorrect (CtoI), so we compute Com=CtoC+ItoC-CtoI as an additional metric. The observations are consistent with findings in our paper. We will add these additional results to the paper. Reviewer mentioned that "the effect of the random initialization and shuffling (RIS) in the algorithm is not clear". We discussed this in lines 284-293 (page 8) where we show that given the same ensemble size $m = 20$ ExtBagging, DynSnap-cyc and DynSnap-step that use RIS outperform Snapshot which does not use RIS. In sensitivity analysis for snapshot number $N$ (results and settings presented in Figure 1c and Appendix J, respectively), where using DynSnap-cyc with $\beta = 1$ i.e. ensemble of $N$ best models without RIS, we show that as $N$ doubles, the performance improves but the improvement is minor after $N = 20$. However, performance of DynSnap-cyc with $\beta = \beta^*$ is further improved partially due to RIS.

**Reviewer #3** points out "the proposed method is not significantly more powerful than the ExtBagging". We discussed this in lines 284-287 (page 8). Our theoretical and empirical results demonstrate that ExtBagging performs very well in both accuracy and consistency as it selects components with the best estimated accuracy and utilizes RIS. Our proposed method DynSnap-cyc combines techniques from ExtBagging and Snapshot to achieve a slightly better performance than ExtBagging but *with substantially reduced training cost*. The reviewer also points that "ACC and CON seem to be correlated". In our experiments, we observe examples like ExtBagging, that have better ACC but lower CON than DynSnap-cyc on CIFAR10+ResNet20 and YAHOO!Answer+fastText. Another example is DynSnap-cyc (ACC 75.64%, CON 85.70%), that has improved accuracy by 3.7% and consistency by 11.3% (significant improvement) compared with Snapshot (ACC 72.96%, CON 76.98%) on CIFAR100+ResNet56. The reviewer also asks "whether the theoretical findings work correctly when any other distance measure is used". Yes, indeed, our theoretical findings are true for Hamming, Euclidean, Manhattan, and Minkowski distances. The proofs can be generalized by using Minkowski inequality 'M.Voitsekhovskii,*Minkowski inequality*,Encyclopedia of Math.,2001' with corresponding order $p$ (replace order 2 in Equation 8 of Appendix B with order $p$). We will provide the generalization proof in supplementary materials. We thank the reviewer for the suggestion "to boldface the highest hits" (we will do this) and "provide more details about replication results" (5 replicates per method).

**Reviewer #4** has asked "why having learning rate scheduling was important". In 'I.Loshchilov, et.al.,*Sgdr: Stochas. grad. desc. with warm restarts*, arXiv preprnt. arXiv:1608.03983,2016', authors suggest that cycling annealing schedule perturbs the parameters of a converged model, which allows the model to find a better local minimum. In 'G.Huang,et.al.,*Snapshot Ensembles: Train 1, get m for free*,arXiv preprnt. arXiv:1704.00109,2017', authors claim that there is a significant diversity in the local minima when visited during each cycle. Our proposed DynSnap-cyc is inspired by their findings. The results show that DynSnap-cyc (using cycling schedule) outperforms DynSnap-step (using step-wise decay). The reviewer also asks "how does the theoretical justification work in a setting where training data distribution changes over successive model generations". Our theoretical findings are invariant with respect to changes in training data distribution since representation (Equation 1) of consistency is invariant as long as $r_t$, $s_{tj}$ and $\tilde{s}_{tj}$ are all represented in a $p$-dimensional space. If $p$ changes, then it is a different problem where consistency loses its meaning. We will add the discussion in the paper.

Table 1: (WAVG) Percentage of correct→incorrect (CtoI), incorrect→correct (ItoC) and CtoC+ItoC-CtoI (Com) predictions for $D_1 \rightarrow D_2$, $D_2 \rightarrow D_3$ and $D_1 \rightarrow D_3$.

| | CIFAR10+ResNet20 | | | | | | | | | CIFAR100+ResNet56 | | | | | | | | | YAHOO!Answers+fastText | | | | | | | | |
| | $D_1 \rightarrow D_2$ | | | $D_2 \rightarrow D_3$ | | | $D_1 \rightarrow D_3$ | | | $D_1 \rightarrow D_2$ | | | $D_2 \rightarrow D_3$ | | | $D_1 \rightarrow D_3$ | | | $D_1 \rightarrow D_2$ | | | $D_2 \rightarrow D_3$ | | | $D_1 \rightarrow D_3$ | | |
| | ItoC | CtoI | Com | ItoC | CtoI | Com | ItoC | CtoI | Com | ItoC | CtoI | Com | ItoC | CtoI | Com | ItoC | CtoI | Com | ItoC | CtoI | Com | ItoC | CtoI | Com | ItoC | CtoI | Com |
|---|---|---|---|---|---|---|---|---|---|---|---|---|---|---|---|---|---|---|---|---|---|---|---|---|---|---|---|
| SingleBase | 6.60 | 5.38 | 80.79 | 7.08 | 6.83 | 79.59 | 5.74 | 4.27 | 82.15 | 11.18 | 8.80 | 59.56 | 9.36 | 9.26 | 59.19 | 11.32 | 8.86 | 59.60 | 2.32 | 2.15 | 60.83 | 5.07 | 2.75 | 62.56 | 5.14 | 2.65 | 62.65 |
| ExtBagging | 3.11 | 2.13 | **87.13** | 4.40 | 3.80 | **86.07** | 3.13 | 1.56 | **88.31** | 6.57 | 4.42 | 70.65 | 4.87 | 4.71 | 70.53 | 6.81 | 4.51 | 70.73 | 1.72 | 1.82 | 61.54 | 4.43 | 2.29 | 63.21 | 4.65 | 2.61 | 62.88 |
| MCDropout | 6.69 | 5.20 | 80.67 | 7.29 | 7.16 | 78.84 | 5.78 | 4.16 | 81.84 | 10.02 | 9.35 | 58.08 | 10.00 | 9.21 | 59.01 | 10.69 | 9.23 | 58.99 | 2.84 | 1.65 | 62.00 | 7.88 | 5.25 | 61.04 | 7.70 | 3.88 | 62.41 |
| Snapshot | 4.89 | 3.11 | 84.98 | 5.89 | 5.38 | 83.22 | 3.91 | 1.62 | 86.98 | 8.46 | 5.92 | 67.66 | 6.32 | 5.64 | 68.63 | 8.91 | 5.68 | 68.59 | 2.05 | 1.78 | 62.07 | 3.35 | 1.56 | 64.07 | 4.48 | 2.42 | 63.21 |
| DynSnap-cyc | 2.84 | 2.11 | 86.36 | 4.47 | 3.71 | 85.51 | 2.78 | 1.29 | 87.93 | 5.74 | 3.35 | **73.19** | 3.58 | 3.90 | **72.32** | 5.60 | 3.54 | **72.69** | 1.38 | 1.05 | **63.47** | 3.46 | 1.67 | **64.63** | 4.08 | 1.97 | **64.34** |
| DynSnap-step | 3.09 | 2.47 | 86.38 | 4.36 | 3.64 | 85.91 | 3.47 | 2.13 | 87.42 | 6.44 | 5.92 | 68.14 | 5.62 | 4.93 | 69.82 | 5.98 | 4.77 | 69.98 | 1.77 | 1.28 | 63.23 | 3.52 | 1.71 | 64.62 | 4.29 | 1.99 | **64.34** |

[Meta-Review · NeurIPS 2020]

The authors define a notion of consistency and correct-consistency for ensembles, and show this can be used for dynamic ensemble pruning. Ensembles considered are generated for neural networks using snapshots taken during learning. I agree with the authors that the main concern of reviewer one, which is the only strong voice for rejection, is a bit too harsh, as the same criticism applies to many if not all other ensemble properties that have been investigated in the past. Ensembles do not in all cases perform better than their single best member (that would probably violate the NFL theorem), but they often do in practise. The paper also introduces a dedicated ensemble learning method for neural networks, based on "extended bagging" ideas as well as cyclical learning rate schedule. The novel addition is a pruning criterion, based on a formula with one parameter beta, where the theory suggests a specific value for beta. At least in the Cifar100 experiment reported, this suggested beta value is shown to perform well. So in summary, taking all the author feedback into account, I suggest accepting this submission.